# What Knowledge Gets Distilled in Knowledge Distillation?

**Utkarsh Ojha**[*]    **Yuheng Li**[*]    **Anirudh Sundara Rajan**[*]

**Yingyu Liang**    **Yong Jae Lee**

University of Wisconsin-Madison

## Abstract

Knowledge distillation aims to transfer useful information from a teacher network to a student network, with the primary goal of improving the student's performance for the task at hand. Over the years, there has a been a deluge of novel techniques and use cases of knowledge distillation. Yet, despite the various improvements, there seems to be a glaring gap in the community's fundamental understanding of the process. Specifically, what is the knowledge that gets distilled in knowledge distillation? In other words, in what ways does the student become similar to the teacher? Does it start to localize objects in the same way? Does it get fooled by the same adversarial samples? Does its data invariance properties become similar? Our work presents a comprehensive study to try to answer these questions. We show that existing methods can indeed indirectly distill these properties beyond improving task performance. We further study why knowledge distillation might work this way, and show that our findings have practical implications as well.

## 1 Introduction

Knowledge distillation, first introduced in [2, 12], is a procedure of training neural networks in which the 'knowledge' of a teacher is transferred to a student. The thesis is that such a transfer (i) is possible and (ii) can help the student learn additional useful representations. The seminal work by [12] demonstrated its effectiveness by making the student imitate the teacher's class probability outputs for an image. This ushered in an era of knowledge distillation algorithms, including those that try to mimic intermediate features of the teacher [26, 33, 13], or preserve the relationship between samples as modeled by the teacher [22, 32], among others.

While thousands of papers have been published on different techniques and ways of using knowledge distillation, there appears to be a gap in our fundamental understanding of it. Yes, it is well-known that the student's performance on the task at hand can be improved with the help of a teacher. But what exactly is the so-called *knowledge* that gets distilled during the knowledge distillation process? For example, does distillation make the student look at similar regions as the teacher when classifying images? If one crafts an adversarial image to fool the teacher, is the student also more prone to getting fooled by it? If the teacher is invariant to a certain change in data, is that invariance also transferred to the student? Such questions have not been thoroughly answered in the existing literature.

This has become particularly relevant because there have been studies which present some surprising findings about the distillation process. [3] showed that performing knowledge distillation with a bigger teacher does not necessarily improve the student's performance over that with a smaller teacher, and thus raised questions about the effectiveness of the distillation procedure in such cases. [29] showed that the agreement between the teacher and distilled student's predictions on test images is

---

[*]Equal contribution

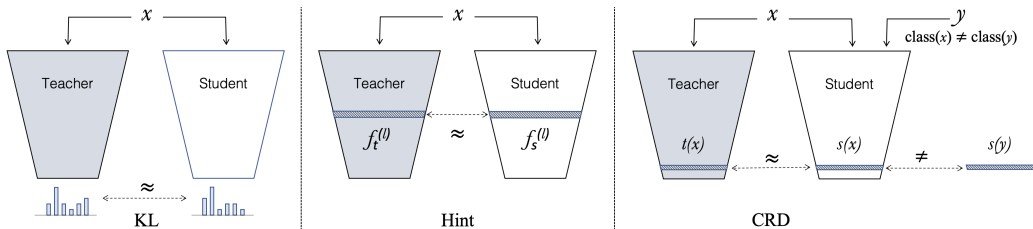

Figure 1: Methods used in this work. (i) $KL$ [12]: mimicking of class probabilities. (ii) $Hint$ [26]: mimicking of features at an intermediate layer. (iii) $CRD$ [30]: features from the student and teacher for the same image constitute a positive pair, and those from different classes make up a negative pair.

not that different to the agreement of those between the teacher and an independently trained student, raising further doubts about how knowledge distillation works, if it works at all.

In this work, we present a comprehensive study tackling the above questions. We analyze three popular knowledge distillation methods [12, 26, 30]. Many of our findings are quite surprising. For example, by simply mimicking the teacher's output using the method of [12], the student can inherit many implicit properties of the teacher. It can gain the adversarial vulnerability that the teacher has. If the teacher is invariant to color, the student also improves its invariance to color. To understand why these properties get transferred without an explicit objective to do so, we study the distillation process through a geometric lens, where we think about the features from a teacher as relative positions of an instance (i.e., distances) from its decision boundary. Mimicking those features, we posit, can therefore help the student inherit the decision boundary and (consequently) the implicit properties of the teacher. We show that these findings have practical implications; e.g., an otherwise *fair* student can inherit biases from an unfair teacher. Hence, by shedding some light on the 'dark knowledge' [11], our goal is to dissect the distillation process better.

## 2 Related work

Model compression [2] first introduced the idea of knowledge distillation by compressing an ensemble of models into a smaller network. [12] took the concept forward for modern deep learning by training the student to mimic the teacher's output probabilities. Some works train the student to be similar to the teacher in the intermediate feature spaces [26, 33]. Others train the student to mimic the relationship between samples produced by the teacher [22, 31, 23], so that if two samples are close/far in the teacher's representation, they remain close/far in the student's representation. Contrastive learning has recently been shown to be an effective distillation objective in [30]. More recently, [1] present practical tips for performing knowledge distillation; e.g., providing the same view of the input to both the teacher and the student, and training the student long enough through distillation. Finally, the benefits of knowledge distillation have been observed even if the teacher and the student have the same architecture [5, 34]. For a more thorough survey of knowledge distillation, see [10]. In this work, we choose to study three state-of-the-art methods, each representative of the output-based, feature-based, and contrastive-based families of distillation approaches.

There have been a few papers that present some surprising results. [3] shows that a smaller & less accurate teacher is often times better than a bigger & more accurate teacher in increasing the distilled student's performance. More recent work shows that the agreement between the predictions of a teacher and student is not necessarily much higher than that between the teacher and an independent student [29]. There has been work done which tries to explain why distillation *improves student's performance* by trying linking it to the regularizing effect of soft labels [19, 35] or what the key ingredients are which help in student's optimization process [24, 14]. What we seek to understand in this work is different: we study different ways (beyond performance improvement) in which a student becomes similar to the teacher by inheriting its implicit properties.

## 3 Distillation methods studied

To ensure that our findings are general and cover a range of distillation techniques, we select standard methods representative of three families of distillation techniques: output-based [12], feature-based [26], and contrastive-based [30]. The objectives of these methods, described below, are combined with the cross entropy loss $\mathcal{L}_{\text{CLS}}(\mathbf{z_s}, \mathbf{y}) := -\sum_{j=1}^{c} y_j \log \sigma_j(\mathbf{z_s})$, where $\mathbf{y}$ is the

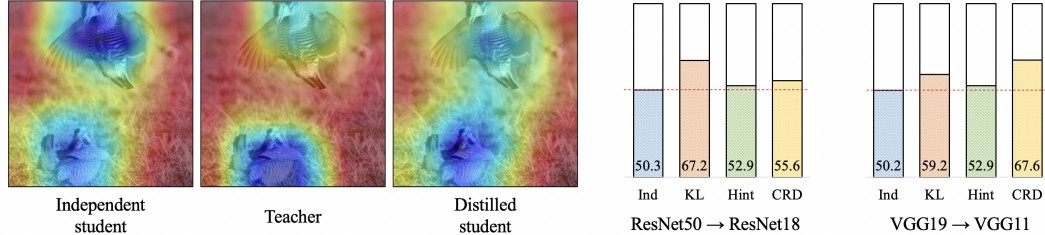

| Independent student | Teacher | Distilled student | ResNet50 → ResNet18 | VGG19 → VGG11 |
|---|---|---|---|---|

Figure 2: **Left:** An example of how the distilled student can focus on similar regions as the teacher while classifying an image. **Right:** % where teacher's CAM is more similar to the distilled student's CAM than to the independent student's CAM. The red line indicates chance performance (50%).

ground-truth one-hot label vector, $\mathbf{z_s}$ is the student's logit output, $\sigma_j(\mathbf{z}) = \exp(z_j)/\sum_i \exp(z_i)$ is the softmax function, and $c$ is the number of classes.

(1) **KL:** [12] proposed to use the soft labels produced by the teacher as an additional target for the student to match, apart from the (hard) ground-truth labels. This is done by minimizing the KL-divergence between the predictive distributions of the student and the teacher:

$$\mathcal{L}_{\mathrm{KL}}(\mathbf{z_s}, \mathbf{z_t}) := -\tau^2 \sum_{j=1}^{c} \sigma_j\left(\frac{\mathbf{z_t}}{\tau}\right) \log \sigma_j\left(\frac{\mathbf{z_s}}{\tau}\right), \tag{1}$$

where $\mathbf{z_t}$ is the logit output of the teacher, and $\tau$ is a scaling temperature. The overall loss function is $\gamma\mathcal{L}_{\mathrm{CLS}} + \alpha\mathcal{L}_{\mathrm{KL}}$, where $\gamma$ and $\alpha$ are balancing parameters. We refer to this method as $KL$.

(2) **Hint:** FitNets [26] makes the student's intermediate features ($\mathbf{f_s}$) mimic those of the teacher's ($\mathbf{f_t}$) for an image $x$, at some layer $l$. It first maps the student's features (with additional parameters $r$) to match the dimensions of the teacher's features, and then minimizes their mean-squared error:

$$\mathcal{L}_{Hint}(\mathbf{f_s^{(1)}}, \mathbf{f_t^{(1)}}) = \frac{1}{2}||\mathbf{f_t^{(1)}} - r(\mathbf{f_s^{(1)}})||^2 \tag{2}$$

The overall loss is $\gamma\mathcal{L}_{\mathrm{CLS}} + \beta\mathcal{L}_{\mathrm{Hint}}$, where $\gamma$ and $\beta$ are balancing parameters. [26] termed the teacher's intermediate representation as $Hint$, and we adopt this name.

(3) **CRD:** Contrastive representation distillation [30] proposed the following. Let $s(x)$ and $t(x)$ be the student's and teacher's penultimate feature representation for an image $x$. If $x$ and $y$ are from different categories, then $s(x)$ and $t(x)$ should be similar (positive pair), and $s(x)$ and $t(y)$ should be dissimilar (negative pair). A key for better performance is drawing a large number of negative samples $N$ for each image, which is done using a contantly updated memory bank.

$$\mathcal{L}_{\mathrm{CRD}} = -\log h(s(x), t(x)) - \sum_{j=1}^{N} \log(1 - h(s(x), t(y_j))) \tag{3}$$

where $h(a, b) = (e^{a \cdot b/\tau})/(e^{a \cdot b/\tau} + \frac{N}{M})$, $M$ is the size of the training data, $\tau$ is a scaling temperature, and $\cdot$ is the dot product. We use $CRD$ to refer to this method. All other implementation details (e.g., temperature for *KL*, layer index for *Hint*) can be found in appendix.

## 4 Experiments

We now discuss our experimental design. To reach conclusions that are generalizable across different architectures and datasets, and robust across independent runs, we experiment with a variety of teacher-student architectures, and tune the hyperparameters so that the distillation objective improves the test performance of the student compared to independent training. For each setting, we average the results over two independent runs. We report the top-1 accuracy of all the models in the appendix. The notation $\mathtt{Net_1} \rightarrow \mathtt{Net_2}$ indicates distilling the knowledge of $\mathtt{Net_1}$ (teacher) into $\mathtt{Net_2}$ (student).

### 4.1 Does localization knowledge get distilled?

We start by studying whether the localization properties of a teacher transfers to a student through knowledge distillation. For example, suppose that the teacher classifies an image as a cat by focusing

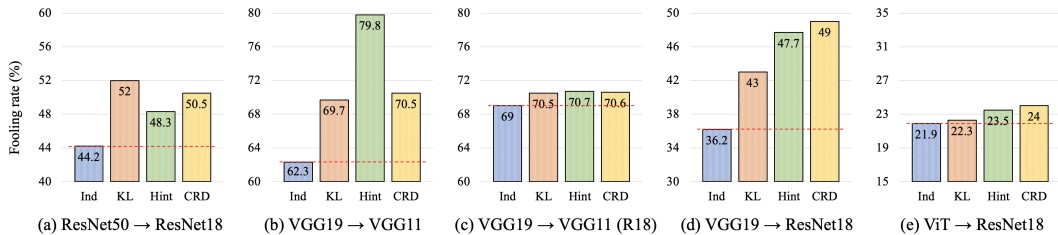

Figure 3: The images which fool the teacher fool the distilled student more than the independent student in **(a)**, **(b)** and **(d)**, but not in **(e)**. If the adversarial attack is generated using a foreign network (ResNet18, **(c)**), it fails to convincingly fool the distilled student more than the independent one.

on its face, and an independent student classifies it as a cat by focusing on its fur. After knowledge distillation, will the student focus *more* on the face when classifying the image as a cat?

**Experimental setup:** We train three models for ImageNet classification: a teacher, a distilled and an independent student. For each network, we obtain the class activation map (CAM) of the ground-truth class for each of random 5000 test images, using Grad-CAM [27], which visualizes the pixels that a classifier focuses on for a given class. We then compute how often (in terms of % of images) the teacher's CAM is more similar (using cosine similarity) to the distilled student's CAM than to the independent student's CAM. A score of >50% means that, on average, the distilled student's CAMs become more similar to those of the teacher than without distillation. As a sanity check, we also compute the same metric between the teacher's CAM and the CAMs of two independent students trained with different seeds (*Ind*), which should be equally distant from the teacher (score of ∼50%).

**Results:** Fig. 2 (right) shows the results for two configurations of teacher-student architectures: (i) ResNet50 → ResNet18 and (ii) VGG19 → VGG11. For *KL* and *CRD*, we observe a significant increase in similarity compared to random chance (as achieved by the *Ind* baseline). *Hint* also shows a consistent increase, although the magnitude is not as large.

**Discussion:** This experiment shows that having access to the teacher's class-probabilities, i.e. confidence for the ground-truth class, can give information on where the teacher is focusing on while making the classification decision. This corroborates, to some degree, the result obtained in the Grad-CAM paper [27], which showed that if the network is very confident of the presence of an object in an image, it focuses on a particular region (Fig. 1(c) in [27]), and when it is much less confident, it focuses on some other region (Fig. 7(d) in [27]). Fig. 2 (left) shows a sample test image and the corresponding CAMs produced by the independent student (left), teacher (middle), and distilled student (right) for the ground-truth class. The distilled student looks at similar regions as the teacher, and moves away from the regions it was looking at initially (independent student). So, regardless of the correctness of any network's CAM, our analysis shows that a distilled student's CAM does become similar to those of the teacher for all three distillation techniques, albeit with varying degrees.

## 4.2 Does adversarial vulnerability get distilled?

Next, we study the transfer of a different kind of property. If we design an adversarial image to fool the teacher, then will that same image fool the distilled student more than the independent student?

**Experimental setup:** We train a teacher, a distilled student, and an independent student for ImageNet classification. Given 5000 random test images, we convert each image $I$ into an adversarial image $I^{adv}$ using iterative FGSM [9, 17] (see appendix for details), whose goal is to fool the teacher, so that teacher's prediction for $I^{adv}$ changes from its original prediction for $I$. Fooling rate is then defined as the fraction of adversarial images which succeed at this task. In our experiments, we use only this fraction of adversarial images which fool the teacher, and apply them to different students.

**Results:** We evaluate four configurations: (i) ResNet50 → ResNet18, (ii) VGG19 → VGG11, (iii) VGG19 → ResNet18, (iv) ViT (ViT-b-32) [4] → ResNet18. The fooling rate is ∼85% for all the teachers. Fig. 3 shows the fooling rate (y-axis) when applying these successful adversarial images to different types of students (x-axis). We see that for ResNet50 → ResNet18, the ability to fool the independent student drops to 44.2%, which is expected since the adversarial images aren't designed for that student. In distilled students, we see an increase in the fooling rate relative to the independent one across all distillation methods (48%-52%). The trend holds for VGG19 → VGG11 and VGG19

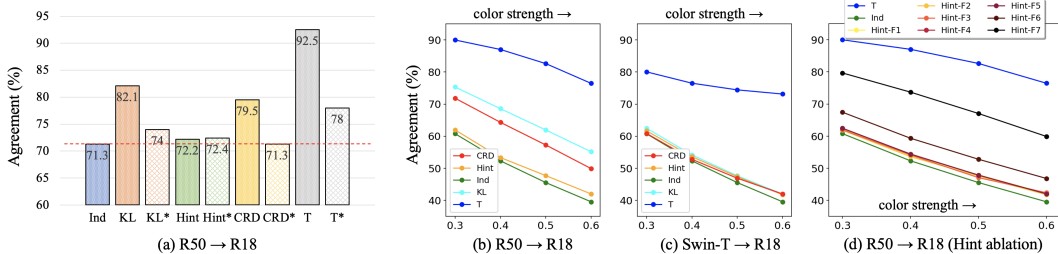

Figure 4: **(a)** Agreement between two images with different color properties. * indicates distillation done by a teacher T* not trained to be color invariant. **(b, c)** Agreement between two images having increasingly different color properties. **(d)** Effect of different layers used in $Hint$ distillation.

→ ResNet18; Fig. 3 (b, d). When distillation is done from a transformer to a CNN (ViT → ResNet18) the fooling rates remain similar for the independent and distilled students; Fig. 3 (e). Here, we don't see the student becoming similar to the teacher to the extent observed for a CNN → CNN distillation.

**Discussion:** This result is surprising. Iterative FGSM is a white box attack, which means that it has full access to the target model (teacher), including its weights and gradients. Thus, the adversarial examples are specifically crafted to fool the teacher. In contrast, the student never has direct access to the weights and gradients of the teacher, regardless of the distillation method. Yet, by simply trying to mimic the teacher's soft probabilities ($KL$) or an intermediate feature layer ($Hint$, $CRD$), the student network inherits, to some extent, the particular way that a teacher is fooled.

We conduct an additional study to ensure that the reason the distilled student is getting fooled more is because it is being attacked specifically by its *teacher's* adversarial images; i.e., if those images are designed for some other network, would the difference in fooling rates still be high? We test this in the VGG19 → VGG11 setting. This time we generate adversarial images ($I → I^{adv}$) to fool an ImageNet pre-trained ResNet18 network, instead of VGG19 (the teacher). We then use those images to attack the same VGG11 students from the VGG19 → VGG11 setting. In Fig. 3 (c), we see that the fooling rates for the independent and distilled students remain similar. This indicates that distillation itself does not make the student more vulnerable to *any* adversarial attack, and instead, an increase in fooling rate can be attributed to the student's inheritance of the teacher's adversarial vulnerability.

### 4.3 Does invariance to data transformations get distilled?

We have studied whether the response properties on single images get transferred from the teacher to the student. Now suppose that the teacher is invariant to certain changes in data, either learned explicitly through data augmentation or implicitly due to architectural choices. Can such properties about *changes in images* get transferred during distillation?

**Experimental setup:** We study color invariance as a transferable property. We train three models for ImageNet classification: a teacher, a distilled and an independent student. While training the teacher, we add color jitter in addition to the standard augmentations (random crops, horizontal flips). Specifically, we alter an image's brightness, contrast, saturation, and hue, with magnitudes sampled uniformly in [0, 0.4] for the first three, and in [0, 0.2] for hue. This way, the teacher gets to see the same image with different color properties during training and can become color invariant. When training the student, we only use the standard augmentations *without* color jittering, and see whether such a distilled student can indirectly inherit the color invariance property through the teacher.

**Results:** We start with the ResNet50 → ResNet18 configuration. After training, we evaluate the models on 50k ImageNet validation images. For each image $X$, we construct its augmented version $X'$ by altering the color properties in the same way as done while training the teacher. Fig. 4 (a) depicts the agreement scores between $X$ and $X'$ (y-axis) for different models (x-axis). The teacher (T), being trained to have that property, achieves a high score of 92.5%. The independent student, which is not trained to be color invariant, has a much lower score of 71.3%. But, when the color-invariant teacher is used to perform distillation using $KL$ or $CRD$, the agreement scores of the students jump up to 82.1% and 79.5% respectively. To ensure that this increase is not due to some regularizing property of the distillation methods that has nothing to do with the teacher, we repeat the experiment, this time with a ResNet50 teacher (T*) that is not trained with color augmentation. The new agreement scores for the distilled students (marked by *; e.g., KL*) drop considerably

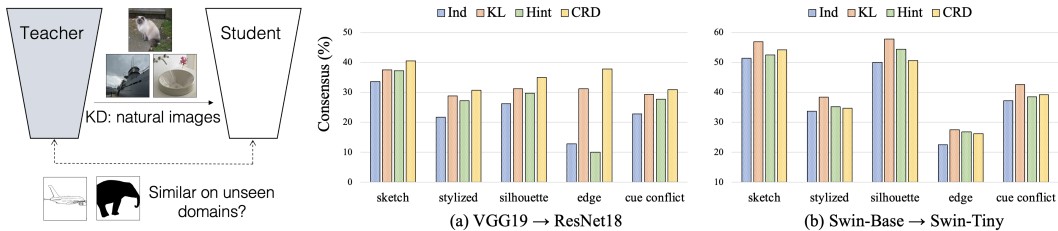

Figure 5: **Left:** Does knowledge transferred about one domain give knowledge about other unseen domains? **Right:** Consensus scores between teacher and student for images from unseen domains.

compared to the previous case. This is a strong indication that the student does inherit *teacher-specific* invariance properties during distillation.

In Fig. 4(b), we show that this trend in agreement scores (y-axis) holds even when the magnitude of change in brightness, contrast and saturation (x-axis) is increased to create $X'$ from $X$. We repeat this experiment for Swin-tiny (a transformer) [20] $\rightarrow$ ResNet18 in Fig. 4(c). We again see some improvements in the agreement scores of the distilled students. This increase, however, is not as significant as that observed in the previous CNN $\rightarrow$ CNN setting. Throughout this work, we find that distilling the properties from a transformer teacher into a CNN student is difficult. The implicit biases introduced due to architectural differences between the teacher (transformer) and student (CNN) seem too big, as was studied in [25], to be overcome by current distillation methods.

We also note the ineffectiveness of $Hint$ in distilling this knowledge. Our guess for this is the choice of $l$ for distillation, which is typically set to be in the middle of the network as opposed to deeper layers where $KL$ and $CRD$ operate. So, we perform an ablation study for $Hint$, where the student mimics a different feature of the teacher each time; starting from a coarse feature of resolution $56 \times 56$ (F1) to the output scores (logits) of the teacher network (F7). We plot the agreement scores in Fig. 4(d), where we see that the score increases as we choose deeper features. Mimicking the deeper layers likely constrains the overall student more compared to a middle layer, since in the latter case, the rest of the student (beyond middle layer) could still function differently compared to the teacher.

**Discussion:** In sum, color invariance can be transferred during distillation. This is quite surprising since the teacher is not used in a way which would expose any of its invariance properties to the student. Remember that all the student has to do during distillation is to match the teacher's features for an image $X$; i.e., the student does not get to see its color augmented version, $X'$, during training. If so, then why should it get to know how the teacher would have responded to $X'$?

### 4.4 Does knowledge about unseen domains get distilled?

So far, our analysis has revolved around the original ImageNet dataset, something that was used to perform the distillation itself. So, does the teacher only transfer its knowledge pertaining to this domain, or also of domains it has never seen (Fig. 5 left)?

**Experimental setup:** To test this, we first perform distillation using two settings (i) VGG19 $\rightarrow$ ResNet18 and (ii) Swin-Base [20] $\rightarrow$ Swin-Tiny, where the training of the teacher, as well as distillation is done on ImageNet. During test time, we take an image from an unseen domain and see how frequently the student's and teacher's class predictions match for it (regardless of whether the predicted label is correct or incorrect), which we call the consensus score. For the unseen domains, we consider the five datasets proposed in [7]: sketch, stylized, edge, silhouette, cue conflict. Images from these domains are originally from ImageNet, but have had their properties modified.

**Results:** Fig. 5 (right) shows the consensus scores between the teacher and the student (y-axis). We see a nearly consistent increase in consensus brought about by the distillation methods in both settings. The extent of this increase, however, differs among the domains. There are cases, for example, in VGG19 $\rightarrow$ ResNet18, where consensus over images from edge domain increases over 100% (12% $\rightarrow \geq 30\%$) by some distillation methods. On the other hand, the increase can also be relatively modest in certain cases: e.g. Swin-Base $\rightarrow$ Swin-Tiny in stylized domain.

**Discussion:** [29] showed that the agreement in classification between the teacher and distilled student is not much different than that to an independent student on CIFAR. We find this to be true in our ImageNet classification experiments as well. That is why the increase in agreement observed for

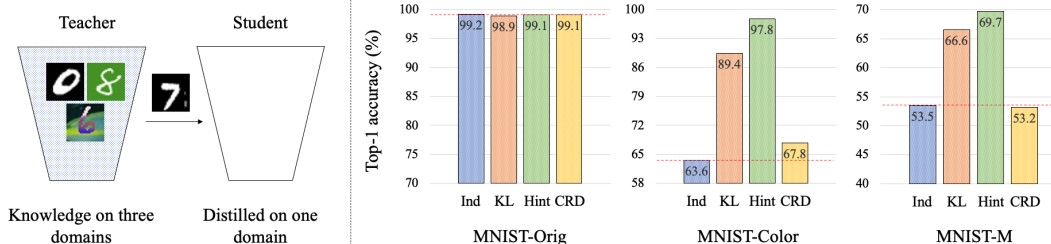

Figure 6: **Left:** The teacher is trained on three domains: `MNIST-Orig`, `MNIST-Color`, and `MNIST-M`. Distillation is done only on `MNIST-Orig`. **Right:** Test accuracy of the students. Note the increase in performance on `MNIST-Color` & `MNIST-M` domains by the distilled students.

unseen domains is surprising. After all, if the agreement is not increasing when images are from the seen domain, why should it increase when they are from an unseen domain? It is possible that there is more scope for increase in agreement in unseen domains vs seen domain. The consensus score between teacher and independent student for the seen domain is $\geq 75\%$ (appendix), whereas for an unseen domain, e.g. `sketch`, it is $\leq 40\%$. So, it might not be that knowledge distillation does not work, as the authors wondered in [29], but its effect could be more prominent in certain situations.

### 4.5  Other studies

In appendix, we study additional aspects of a model such as shape/texture bias, invariance to random crops, and find that even these obscure properties can transfer from a teacher to the student. We also explore the following questions: If we can find alternative ways of increasing a student's performance (e.g. using crafted soft labels), will that student gain similar knowledge as a distilled student? If distillation cannot increase the student's performance, is there no knowledge transferred? Finally, we also present results on some more datasets like CIFAR100 [16], VLCS and PACS [18], showing that the phenomena of the implicit transfer of properties dueing knowledge distillation extends even to datasets beyond ImageNet and MNIST.

## 5  Applications

Beyond the exploratory angle of studying whether certain properties get transferred during distillation, the idea of a student becoming similar to a teacher in a broad sense has practical implications. We discuss a good and a bad example in this section.

### 5.1  The Good: the free ability of domain adaptation

Consider the following setup: the teacher is trained for a task by observing data from multiple domains ($\mathcal{D}_1 \cup \mathcal{D}_2$). It is then used to distill knowledge into a student on only $\mathcal{D}_1$. Apart from getting knowledge about $\mathcal{D}_1$, will the student also get knowledge about $\mathcal{D}_2$ indirectly?

**Experimental setup:** We use MNIST digit recognition, and train the teacher on three domains: (i) `MNIST-orig`: original gray-scale images from MNIST, (ii) `MNIST-Color`: background of each image randomly colored, and (iii) `MNIST-M` [6]: MNIST digits pasted on random natural image patches. The student models are trained *only* on `MNIST-orig` and evaluated (top-1 accuracy) on all three domains. The network architecture is same for both the teacher and the student (see appendix).

**Results:** When the independent student is trained only on `MNIST-orig`, its performance drops on the unseen domains, which is expected due to domain shift. The distilled students (especially $KL$ & $Hint$), however, are able to significantly improve their performance on both unseen domains; Fig. 6.

**Discussion:** This result shows distillation's practical benefits: once a teacher acquires an ability through computationally intensive training (e.g., training on multiple datasets), that ability can be distilled into a student, to a decent extent, through a much simpler process. The student sees the teacher's response to *gray-scale* images (`MNIST-orig`) that lack any color information. But that information helps the student to deal better with *colored* images (e.g., `MNIST-Color`), likely because the teacher has learned a domain-invariant representation (e.g., shape) which is distilled to the student.

## 5.2 The Bad: Students can inherit harmful biases from the teacher

Consider the problem of classifying gender from human faces. Imagine an independent student which performs the classification fairly across all races. The teacher, on the other hand, is biased against certain races, but is more accurate than the student *on average*. Will the student, which was originally fair, become unfair after mimicking the unfair teacher?

**Experimental setup:** We consider a ResNet20 $\rightarrow$ ResNet20 setting, and use FairFace dataset [15], which contains images of human faces from 7 different races with their gender labeled. From its training split, we create two different subsets ($\mathcal{D}_s$ and $\mathcal{D}_t$) with the following objectives - (i) $\mathcal{D}_s$ has a particular racial composition so that a model trained on it will perform fairly across all races during test time; (ii) $\mathcal{D}_t$'s composition is intended to make the model perform unfairly for certain races. The exact composition of $\mathcal{D}_s$ and $\mathcal{D}_t$ is given in the appendix. The teacher is trained on $\mathcal{D}_t$ whereas the independent/distilled students are trained on $\mathcal{D}_s$. We use $KL$ for distillation.

**Results:** We observe in the figure on the right that the independent student performs roughly fairly across all races, which is what we intended. The teacher is more accurate than the student, but performs relatively poorly on faces from Race 3 compared to others. After distillation, we observe that the student's *overall* accuracy improves, but the gain in accuracy is less for images from Race 3. So, when it is trained on $\mathcal{D}_s$ by itself, it behaves fairly. But when it mimics the teacher on $\mathcal{D}_s$, it becomes unfair.

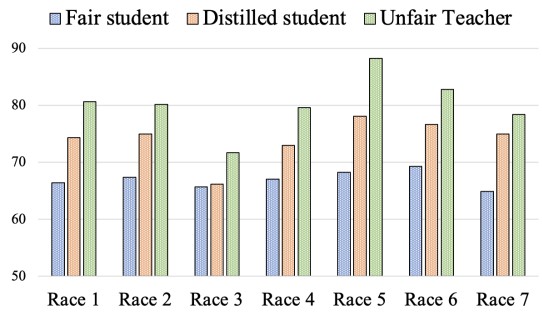

This follows the observation made in [21], where the authors found that the performance gains during distillation might not be spread uniformly across different sub-categories.

**Discussion:** The practical takeaway from the experiment is that knowledge distillation can bring forth behaviour which is considered socially problematic if it is viewed simply as a blackbox tool to increase a student's performance on test data, as it did in the previous example. Proper care must hence be taken regarding the transfer/amplification of unwanted biases in the student.

## 6 Why does knowledge distillation work in this way?

**An Illustrative Example.** Why should a teacher's response to an image contain such rich information about its implicit properties? We first intuitively explain this through a toy classification problem using $KL$ as the distillation objective. Fig. 7 (left) shows data points from two classes (red and blue). The teacher has access to the complete set, thereby learning to classify them appropriately (orange decision boundary). On the other hand, suppose that the training of the independent and distilled student is done on a subset of the points. The independent student can learn to classify them in different ways, where each decision boundary can be thought of as looking for different features, hence giving rise to different properties. But the distilled student is trying to mimic the teacher's class probabilities, which reflect the distance from the (orange) decision boundary of the teacher, visualized as circles around the points. Then the decision boundary of the distilled student should remain tangential to these circles so that it preserves the teacher's distances. So, this very simple case depicts that mimicking the class probabilities can force the student's decision boundary to resemble that of the teacher, which is *not* true for the independent student.

A mathematical formulation of the same case can be considered as follows. Let $D$ be a linearly separable dataset: $D \in \{(x_1, y_1), (x_2, y_2), ......, (x_n, y_n) | x_i \in \Re^m, y_i \in \{-1, 1\}\}$. For an arbitrary linear classifier having weights $W \in \Re^{m \times 1}$ and biases $b \in \Re$, the binary classification objective for the teacher and the independent student can be described as $\forall(x_i, y_i) \in D, (W^T x_i + b)y_i > 0$.

Since the dataset is linearly separable, both the teacher as well as the independent student can solve this system of inequalities in many ways. Let $\{W_t, b_t\}$ be the parameters of one particular teacher.

Next, we look at the objective of the distilled student, who tries to map $x_i$ to $z_i = W_t^T x_i + b_t$, instead of $y_i$. That is, the optimal solution satisfies: $\forall(x_i, z_i) \in D_{dist}, W_s^T x_i + b_s = z_i$. This system of linear equations will have a unique solution provided it has access to at least $m + 1$ training instances and that these instances are linearly independent. Since we know that the teacher's weights and

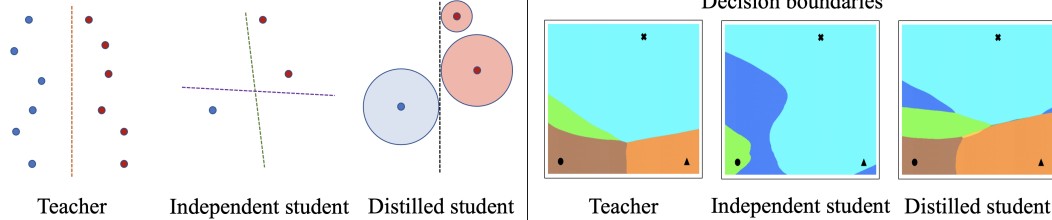

Figure 7: **Left.** Teacher learns on the the entire dataset. Independent student without access to the full dataset might learn some spurious correlations. Distillation provides constraints to reconstruct the teacher's decision boundary. **Right.** Sampled decision boundaries for points from MNIST-Color visualized using [28]. The distilled student is better at reconstructing the teacher's decision boundary.

biases, $\{W_t, b_t\}$, can satisfy this equation, we can conclude that if there exists a unique solution, then $W_s = W_t$ and $b_s = b_t$. So, the distilled student, when solving the distillation objective, will recover the weights and biases of the teacher, and consequently its decision boundary.

**Complex neural networks:** Given that the distances of data points, i.e., decision boundary, can be preserved by the distilled student in the very simple case presented above, we now study whether they can be preserved in a more complex setting involving multi-layer neural networks as well.

There have been works [8, 28] which have studied this for the $KL$ objective: if the teacher and the students are all trained on the same dataset $\mathcal{D}$, the decision boundary of the teacher will be more similar to the distilled student than the independent student *for points from $\mathcal{D}$*. And while this result is useful, it is not clear whether (i) this can help explain why the student inherits teacher's knowledge about other domains, and (ii) whether this holds true for distillation objectives other than $KL$.

To study this, we revisit the setup of Sec. 5.1, where the teacher was trained on three domains and distillation was performed on one (Fig. 6 left). We study what happens to the decision boundary similarity between the students and the teacher on all three domains. To compute that similarity between two neural networks, we use the method introduced in [28]. We randomly sample a triplet of data points from a given domain and create a plane that passes over that triplet. The networks that are being compared receive the same dataset of points in this plane, and we compute intersection over union between their predictions. We report the score averaged over 50 triplets.

|            | **Ind** | **KL** | **Hint** | **CRD** |
|------------|---------|--------|----------|---------|
| MNIST-Orig | 0.9107  | 0.9341 | 0.9317   | 0.9062  |
| MNIST-Color| 0.4156  | 0.6711 | 0.7980   | 0.4733  |
| MNIST-M    | 0.4691  | 0.5872 | 0.6541   | 0.4862  |

Table 1: Decision boundary similarity calculated between students and the teacher.

**Results:** Table 1 shows the similarity scores, where we see that the distilled students' decision boundary are almost always more similar the teacher's than the independent student's. This is particularly evident when the scores are computed for domains not seen by the students; e.g., for `MNIST-Color`, the similarity score increases from 0.416 to 0.671 for the student distilled with $KL$.

**Discussion:** First, we can draw an analogy of this experimental setup with the toy example discussed before. The three `MNIST` related domains (for training the teacher) are similar to the *overall* set of blue+red points used to train the toy teacher (Fig. 7 left). The singular `MNIST-orig` domain (for training the students) is similar to the three points available to train the toy students. Now, what the results from Table 1 show is that the data points available for distillation can help determine the teacher's decision boundary even for points which the student did not have access to. So, similar to how the decision boundary estimated by the toy student can correctly estimate the distances of missing blue/red points, the neural network based student can also estimate the distances of points from unseen domains, e.g., `MNIST-color`, in a similar way as its teacher, thereby inheriting the teacher's behavior on these points. An example of this is given in Fig. 7 (right).

## 7 Limitations

While we have tried to study the distillation process in many settings, there do exist many more which remain unexplored. This is because the analysis conducted in our work has many axes of generalization: whether these conclusions hold (i) for other datasets (e.g. SUN scene classification dataset), (ii) in other kinds of tasks (object detection, semantic segmentation), (iii) in other kinds of architectures (modern CNNs, like ResNext), (iv) or for more recent distillation objectives, etc. Moreover, in Sec. 3.5 in the supplementary, we explore the transferability of shape/texture bias from teacher to student, and find some discrepancy in different distillation objective's abilities. Therefore, the conclusions drawn from our work will be more helpful if one can study all the combinations of these factors. Due to limitations in terms of space and computational requirements, however, we have not been able to study all of those combinations.

## 8 Conclusion

Knowledge distillation is a beautiful concept, but its success i.e, increase in student's accuracy, has often been explained by a transfer of *dark knowledge* from the teacher to the student. In this work, we have tried to shed some light on this dark knowledge. There are, however, additional open questions, which we did not tackle in this work: given the architectures of the teacher and student, is there a limit on how much knowledge can be transferred (e.g., issues with ViT $\rightarrow$ CNN)? If one wants to actively avoid transferring a certain property of the teacher into a student (Sec. 5.2), but wants to distill other useful properties, can we design an algorithm tailored for that? We hope this work also motivates other forms of investigation to have an even better understanding of the distillation process.

## Acknowledgement

This work was supported in part by NSF CAREER IIS2150012, and Institute of Information & communications Technology Planning & Evaluation(IITP) grant funded by the Korea government(MSIT) (No. 2022-0-00871, Development of AI Autonomy and Knowledge Enhancement for AI Agent Collaboration), Air Force Grant FA9550-18-1-0166, the National Science Foundation (NSF) Grants 2008559-IIS, 2023239-DMS, and CCF-2046710.

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
