# OpenReview forum: "What Knowledge Gets Distilled in Knowledge Distillation?"
_NeurIPS.cc/2023/Conference — NeurIPS 2023 poster_

### Official Review · Reviewer_8Y8m · 2023-06-15

**Soundness:** 3 good
**Presentation:** 4 excellent
**Contribution:** 3 good
**Rating:** 7
**Confidence:** 3

**Summary:**

This paper argues that distillation implicitly transfers many different behaviors from the teacher to the student, even though these are not explicitly matched via the distillation objective. These include:
1. The saliency map computed via Grad-CAM
2. Specific adversarial examples of the teacher
3. Invariance that the teacher has learned via data augmentations on color jitter, bridghtness, contrast etc.,
4. Performance on Out-of-Distribution (OoD) data

The distillation approaches studied here are:
- the standard K-divergence based loss,
- the Hint loss which matches embeddings and
- contrastive representation KD

The architectures covered are VGG, ResNet, ViT, and Swin (although not all these architectures are used in all experiments).

All main experiments are on ImageNet. A few additional experiments are on MNIST.

The paper also provides an intuitive argument for why such implicit knowledge is transferred: the distilled student learns the specific decision boundary of the teacher. This is verified for an MNIST setting where the network via some existing techniques for comparing decision boundaries.

The paper argues that these findings have practical implications in (a) allowing the student to generalizing to new domains, (b) but also in transferring harmful biases to the student.

**Strengths:**

1. The questions explored in the paper are novel and interesting. This is a new perspective on what type of knowledge may be inadvertently transferred from the teacher to the student.
2. The paper covers various types of knowledge (namely, saliency maps, OOD, adversarial examples, invariances) and there is clear evidence across the board that the student typically imitates the teacher non-trivially.
3. The paper covers three fundamentally different types of distillation losses, which strengthens their claim.
4. The experiments are designed carefully. In all cases, an "independent" student is considered as a baseline so as to judge whether the distilled-student is indeed copying the teacher.

**Weaknesses:**

1. My main concern is that all the main experiments appear to be limited to one (ImageNet) dataset. While this is certainly a non-trivial and sufficiently realistic dataset, one should be wary of drawing observations from a single dataset. Either a language dataset or another image dataset (say, CIFAR100) would have been ideal. But I understand that it would be hard to add these during the rebuttal.



2.  The argument for "why does KD transfer such implicit knowledge" is not compelling and appears to be somewhat circular. The argument given is that "KD transfers various implicit behaviors of the teacher because the student inherits the decision boundary". The key puzzle however is why the KD objective is able transfer all of the decision boundary, and that too on OoD and adversarial regions the student isn't trained on. Overall, I find the claim that the paper's argument "explains" the phenomenon to be too strong.



### Summary

Overall, this paper presents is an interesting collection of observations, identifying an underlying pattern across various distillation algorithms. Although I have some concerns with (a) the specificity to ImageNet and (b) the paper's claim at explaining these phenomena, I believe future work would definitely find this useful.

### Post-rebuttal update

Increasing the score by 1 to acknowledge the authors' new experiments on adversarial vulnerability and domain adaptation that go beyond ImageNet.

**Questions:**

1. Are there any types of knowledge that _do not_ get transferred? The main paper would benefit from a more balanced and scientific discussion of any such types knowledge.

2. One of the implications laid out in the paper, namely that the teacher may transfer biases to the student, may have also been captured in  other work, which may have been prior to this paper [1].

[1] _Teacher's pet: understanding and mitigating biases in distillation_, Michal Lukasik, Srinadh Bhojanapalli, Aditya Krishna Menon, Sanjiv Kumar https://arxiv.org/abs/2106.10494

**Limitations:**

I couldn't find a limitation section in the paper. I'd appreciate explicitly listing out some limitations e.g., there is lack of clarity on
1. how this generalizes to other datasets
2. why these types of knowledge get transferred
3. what types of knowledge do/do not get transferred

---

> ### Author Rebuttal · Authors · 2023-08-09
>
> **1. All the main experiments appear to be limited to one (ImageNet) dataset. While this is certainly a non-trivial and sufficiently realistic dataset, one should be wary of drawing observations from a single dataset.**
>
> We appreciate that the reviewer recognizes that ImageNet is a non-trivial dataset. But there is another point that we wish to discuss about ImageNet before we present results on some other datasets. Please refer to our response to reviewer **B1Ad**, where we explain our reasoning, and also present some results on new datasets (CIFAR, VLCS and PACS).
>
> **2. The argument for "why does KD transfer such implicit knowledge" is not compelling and appears to be somewhat circular.**
>
> We thank the reviewer for raising an interesting point worth discussion. We will try to expand on our explanation given in Sec. 6 of the paper. The theory of decision boundary transfer (for unseen points as well) depends highly on the points that we are using to perform the distillation during training. For example, If we take the toy example of Fig. 7 (left), the reason the decision boundary could be replicated simply using three points is that those points uniquely define a decision boundary, which is the decision boundary used by the teacher. If we had used some other three points (e.g., collinear points), that precise recovery might not have been possible. In practice, this means that if you had performed distillation on random noise images, the student will not inherit the knowledge from the teacher. All of this is to say that the reason the decision boundary of the OOD points can get distilled is because of the certain points we have used during training, and it wouldn’t work with any arbitrary points. The MNIST data points used for distillation seem to be diverse/appropriate enough to work similarly to the three points in the toy example, and the other unseen domains can pictorially be considered to be the missing points in the figure.
>
> We'll tone down on the 'explains' part a bit as suggested, and also incorporate the above explanation.
>
> **3. Any knowledge which does not get transferred?**
>
> Yes, please refer to Sec. 3.5 in the appendix which shows that it is somewhat difficult to distill the shape bias of the teacher into a texture-biased student, at least in most of the distillation settings. Furthermore, as we mention in lines 197-198, distilling any kind of property from a transformer into a CNN is difficult (e.g., failure to transfer color invariance into a student).
>
> **4. Discussion of related work about bias transfer into a student, and adding a limitation section.**
>
> We thank the reviewer for mentioning the work. The key result from Lukasik et al. is that after the distillation process, the resulting student can have an uneven improvement (or reduction) in performance, which can be linked to certain properties of the teacher. And while the results from Sec. 5.2 in our paper also are surrounding the same theme, we have focused on the fairness of the student; that is, with the same dataset, the student would have remained fair if it was trained by itself, but by mimicking the (unfair) teacher on the same dataset, it now becomes unfair (lines 288-290). We will include a discussion of this work in the camera ready version, and will also include a section detailing the limitations of our work, addressing the points mentioned by the reviewer.

---

> > ### Comment · Reviewer_8Y8m · 2023-08-12
> > **Thanks for the response!**
> >
> > Thanks for your rebuttal! I'm raising my score by 1 point to acknowledge your new experiments.
> >
> > Reg. 2: I still believe that this discussion is still circular in the non-linear neural network) regime. In this regime, it is not clear how many points one would need to uniquely define a decision boundary. Your experiments merely show that the decision boundary somehow gets transferred, but it does not shed insight on the "how?".  So to re-emphasize, I think this section would be much more interesting and rigorous, if the "explanation" claim is removed, and if an open "how?" question was posed.
> >
> > Reg. 3: I'd highly encourage bringing this section in the main paper as it presents a more balanced view of what distillation does or does not do.

---

> > > ### Author Response · Authors · 2023-08-18
> > > **Thank you for the updated rating**
> > >
> > > We thank the reviewer for their updated rating.
> > >
> > > Regarding point #2, while we agree that we have not *proved* why decision boundaries transfer in complex non-linear models (neural networks) for both in-distribution and out-of-distribution points, our goal was to give a geometric intuition of why such a thing would be even possible (for linear models). But, we do agree with the reviewer on their latter point, and as mentioned in our initial response, we will tone down the explanation as a (formal) proof, and talk about it as an another interesting fruit of the distillation process.
> > >
> > > Lastly, we will incorporate #3 in our final version.

---

### Official Review · Reviewer_r3A2 · 2023-06-25

**Soundness:** 2 fair
**Presentation:** 4 excellent
**Contribution:** 3 good
**Rating:** 6
**Confidence:** 3

**Summary:**

The paper presents an empirical study to analyze inner workings of knowledge distillation (KD) methods. The authors try to understand the nature of the knowledge "distilled" from a teacher to a student by studying how various properties get transferred in this learning. The properties include (a) localization information, (b) adversarial vulnerability, (c) invariance to data transformations and (d) domain adaptation. There is further analysis about some other properties in the supplement. The paper analyzes three broad distillation methods on ImageNet with multiple CNN and transformer architectures. Lastly, the authors present positive/negative applications of this understanding for specific tasks (domain adaptation, bias transfer).

**Strengths:**

Strengths:

(1) The paper is nicely organized and very well written. The motivations to study the topic are clear. To my knowledge, this analysis is quite original.

(2) The authors investigate some relevant and interesting directions in a modular way to study KD.

(3) Multiple different architectures and distillation techniques are carefully analyzed for various properties and generally demonstrate consistent results.


**Weaknesses:**

Weaknesses:

The weaknesses I see stem from the purely empirical nature of the study.

(1) The generalizability of results to other types of datasets, tasks, domains remains unknown. (Major)

(2) The selection of properties studied seems a little arbitrary. (Minor)

(3) There is lack of any theoretical analysis to characterize the distilled knowledge. (Minor)

Overall, I like the paper but can't favour stronger acceptance yet. I enjoyed reading through it. It is structured well, the experiments are quite thorough (except in one aspect!) and give some interesting insights about KD. Among the weaknesses I highlighted, I won't hold the last two weaknesses strongly against the paper as (3) is a difficult topic in itself and the studied set of properties could still be regarded as reasonably comprehensive in terms of popular ML topics today. However, considering the purely empirical nature, weakness (1) is major for me that should be tackled before I can more clearly favor acceptance.

**Questions:**

Questions:

(1) **Other datasets/tasks**: How well do the results generalize over other datasets and different tasks (small/large scale images, fine-grained recognition, scene classification etc.). I understand that the number of possible experiments is too large and the paper can't cover everything. However, the same way the paper deserves credit for coverage in terms of distillation methods and architecture selection of teacher and student, it also needs to confirm/study if these results hold for other settings. At least a couple more settings should be included. This is specially important as there are some non-intuitive results and demand further analysis to be confirmed. Also, it would be great if you could illustrate domain adaptation application for some non-MNIST data too.

(2) **Properties selected**: While I am assuming the current set of properties were selected as these are some popular topics of study in ML, I would like to know your rationale behind selecting them. For example did you also consider assessing if the uncertainty of teacher gets transferred to student?

(3) **More questions about experiments**: Does a mixture of simultaneous KL + Hint or other mixtures provide stronger distillation of the properties? Is the localization transfer for transformer -> CNN poor? You could use some other saliency map as localization proxy if grad-CAM is not suitable.

**Limitations:**

The authors mention "We do not have a separate section for limitation, but since ours is an analysis paper, we are discussing the open questions all throughout the paper." While I understand their reasoning and thought process, I also believe they should summarize the limtiations of their experimental setup in the conclusion or a separate section. It is an experimental study and it can't possibly conduct all experiments as part of it. The open questions raised throughout the paper I believe or more related to depth of the analysis. The authors should certainly add the limitations regarding breadth of their analysis.

---

> ### Author Rebuttal · Authors · 2023-08-09
>
> **1. Generalization over other datasets and different tasks**
>
> While we agree that results on other datasets can help bolster our conclusions (more on that later), we want to first clarify why we believe ImageNet is a comprehensive test bed for the kind of properties we have wanted to study in this work. Please take a look at our response to reviewer **B1Ad**, where we explain our reasoning, and also present the results on some of the new datasets (CIFAR, VLCS and PACS).
>
> **2. Rationale behind selecting the particular properties**
>
> Our goal was to study “any” property which is not explicitly encoded in the objective function. Conversely, for example, if a distillation objective was trying to make the class activation map of the student similar to the teacher, then studying CAM similarity (as in Sec. 4.1) will not be as interesting, since one would expect the CAM to become similar.
>
> Among those potential properties (not encoded in the objective function), as the reviewer suggested, we did end up going with the ones that have attracted attention in the ML community. In particular, we wanted to study those properties which, based on our natural thinking, should have remained hidden from the student. For example, as we mention in lines 155-157, why should access to the teacher's class probability output for an image tell us how it gets fooled by an adversarial image?
>
> On a practical end (Sec. 5), we wanted to study those properties which will likely have a real-life application.
>
> Overall, our goal is not to say that these are the properties which are important which should be  studied; but more to show that there can be much more going on in the backend of a distillation process than simply improving a student’s accuracy. And it is very much possible that there can be some even more interesting properties that we might have missed, but something that other people can study being motivated by our work.
>
> **3. Mixture of distillation objectives**
>
> Due to time constraints, we were only able to try one setting, where we combined CRD and KL objective in the case of WRN-40-2 -> ShuffleNetV1 distillation on CIFAR-100 dataset, and studied the transfer of adversarial vulnerability from a teacher into the student. The table below shows the results, where we see that using a combination of KL + CRD objectives for distillation results in better knowledge transfer, compared to using them individually. We will investigate this further for other experiments in our work in the camera ready version. But what this experiment tells us is that the knowledge extracted by one distillation objective (e.g., CRD) is not a superset of knowledge extracted by a different distillation objective; there is something new that each distillation objective adds to the inherited knowledge. We thank the reviewer for asking for this interesting experiment.
>
> |  | WRN 40-2 &rarr; ShuffleNetV1|
> |-----------|:-----------:|
> | Independent | 31.23 |
> | KL | 48.62 |
> | CRD | 49.41|
> | KL + CRD | 52.49|
>
> **4. Adding a section of limitation**
>
> We thank the reviewer for pointing this out, and agree with the recommendation. The analysis conducted in our work has many different axes of generalization: generalization to other datasets (e.g. SUN scene classification dataset), generalization to other kinds of tasks (object detection, semantic segmentation), generalization to other kinds of architectures (modern CNNs, like ResNext), generalization to more recent distillation objectives. In theory, the conclusions will be more helpful if one can study all the combinations of these factors, and hopefully discover a common pattern between them. Due to limitations in terms of space and computational requirements, however, we have not been able to study all of those combinations. Our goal in this work has simply been to show the possibility of the transferability of many properties from a teacher into a student network.

---

> > ### Comment · Reviewer_r3A2 · 2023-08-12
> > **Rebuttal acknowledgement**
> >
> > I want to thank the authors for the rebuttal. The additional experiments strengthen the support for their observations and expand the coverage of analysis, which was much needed. They certainly address my concerns to an extent that I won't consider them as major anymore. Consequently, I am increasing my score from **5 to 6**, assuming incorporation of additional modifications the authors have committed to.

---

> > > ### Author Response · Authors · 2023-08-12
> > > **Response to reviewer**
> > >
> > > We thank the reviewer for their updated score. We will make sure to incorporate the mentioned modifications.

---

### Official Review · Reviewer_B1Ad · 2023-07-06

**Soundness:** 3 good
**Presentation:** 3 good
**Contribution:** 3 good
**Rating:** 6
**Confidence:** 4

**Summary:**

The paper explores different ways in which a student network can inherit implicit properties from a teacher network beyond just improving task performance. The authors study various distillation methods, including output-based, feature-based, and contrastive-based techniques, and show that these methods can indirectly distill properties. The paper also investigates why knowledge distillation works and discusses its practical implications. Overall, the paper aims to shed light on the "dark knowledge" of knowledge distillation and motivate further investigation into the distillation process.

**Strengths:**

1. This paper sheds light on the dark knowledge of knowledge distillation (KD) by investigating a series of questions in KD and gives interesting findings.
2. Each sub-section in experiments is well-organized, and the question is answered with clearly designed experiments and supportive results, followed by insightful discussions.
3. The paper is well-written and the presentation is decent.

**Weaknesses:**

1. The major weakness of this paper is the setup appears to be more exploratory in nature. While exploratory experiments can certainly provide valuable insights and help to understand the findings, they do not substitute for a thorough evaluation that generalizes across diverse datasets. In this paper, the use of only a few datasets may not present a comprehensive picture and could yield misleading results.
2. The selection of benchmark methods in this paper, while representative of some standard approaches, is notably restricted. The experimental comparison lacks several recent knowledge distillation (KD) methods, leaving the evaluation somewhat incomplete. The inclusion of these recent advances in KD techniques would not only enhance the comprehensiveness of the evaluation but also deepen our understanding of the KD process.
(Ren et al., 2022) Better Supervisory Signals by Observing Learning Paths.
(Chen et al., 2021) Distilling Knowledge via Knowledge Review.
(Zhao et al., 2022) Decoupled Knowledge Distillation.

**Questions:**

See above.

**Limitations:**

See above.

---

> ### Author Rebuttal · Authors · 2023-08-09
>
> We thank the reviewer for their comments about the thoroughness of the experiments.
>
>
> The reason we chose a single dataset, ImageNet, for most of our experiments is because it is a very large scale dataset of relatively high resolution images. Its ubiquity has led to many different kinds of architectures being developed to be trained on it: from CNN variants (ResNet/VGGs) to transformer variants (ViT/Swin). Because of this, the computer vision/machine learning community has often been interested in the properties of neural networks trained on ImageNet; e.g., whether the network is shape/texture biased [1], how robust is it to adversarial perturbations [2], or whether it generalizes to unseen domains, interpretability of those models [3]. Hence, if one is interested in studying many different aspects of “knowledge” being distilled into a student, it is more conceivable to do so in an ImageNet setting.
>
> That being said, we do agree that having some results on more datasets will help strengthen the conclusions. Accordingly, we conducted the adversarial vulnerability experiment on CIFAR-100 (section 4.2 in the main paper). We report the results on three different teacher-student settings: (i) Wide ResNet 40-2 (WRN-40-2) -> ShuffleNetV1,  (ii) VGG13 -> VGG8, (iii) ResNet50 -> MobileNetV2. Both the teacher and the students are trained on the training split of CIFAR-100, and tested on 5000 random images from the test split. The results shown below depict the fooling rates (in %) of different kinds of students when using adversarial images crafted for the teacher. We see that the fooling rate increases for distilled students, following a similar pattern as in Sec. 4.2 - Fig. 3. So, adversarial vulnerability of the teacher does get distilled into the students trained with different distillation objectives.
>
> |  | WRN 40-2 &rarr; ShuffleNetV1| VGG-13 &rarr; VGG-8 | ResNet50 &rarr; MobileNetV2 |
> |-----------|:-----------:|:-----------:|:-----------:|
> | Independent | 31.23 | 42.42 | 36.57 |
> | KL | 48.62 | 51.87 | 43.32 |
> | Hint | 62.63| 49.79 | 43.91 |
> | CRD | 49.41| 54.68 | 46.07 |
> | DKD | 57.19| 60.01 | 47.03 |
>
> Furthermore, we also follow the reviewer’s suggestion to report the results on a more recent distillation objective. Specifically, of the works mentioned by the reviewer, we consider the most recent one - Decoupled Knowledge Distillation (DKD) (Zhao et al. 2022) - and present how well it transfers this knowledge, in the last row. We see that the conclusions hold even for this objective. In fact, in two out of the three settings, DKD transfers the adversarial vulnerability the most compared to the other distillation objectives. This gives a further sign that the various ways in which the community has tried to distill the knowledge from a teacher into a student does more than improve the student’s performance. In our camera ready, we will try to include the analysis with this new distillation objective for all the experiments in the paper.
>
> Next, similar to the MNIST domain adaptation experiment in Section 5.1, we conduct the experiment on a more real-world domain. We consider two datasets: VLCS and PACS [4]. VLCS
> consists of images from four domains - VOC2007, LabelMe, Caltech-101, and SUN - where in each domain there are images belonging to five categories. PACS consists of images from four domains as well - sketch, photo, cartoon, art painting. Each domain consists of the same seven object categories. The teacher is trained on all the four domains, but the students (independent and distilled) are trained on three domains (images from one domain are never shown). The unseen domains are Caltech 101 and Photo when working with VLCS and PACS datasets respectively.
>
> The goal is to see if mimicking the teacher on three domains also helps the student inherit teacher’s information on the fourth (hidden) domain; i.e., do we see an improvement in the accuracy on that hidden domain for the distilled students, compared to an independent student. The results, depicting the classification accuracy of different models, are shown below. The distilled students’ performance improves on the unseen domains in both the cases, simply by having access to teacher’s responses on the other three domains. This is particularly pronounced when distillation is done using KL. For example, in VLCS, the performance on the unseen domain (Caltech 101) improves from 54.31 by the independent student to 71.83 by the KL distilled student.
>
> | *VLCS* | Caltech 101 (unseen) | LabelMe (seen) | SUN09 (seen) | VOC 2007 (seen) |
> |-----------|:-----------:|:-----------:|:-----------:|:-----------:|
> | Independent | 54.31|	61.07|	60.38|	51.94 |
> | KL | 71.83|	60.73|	61.52|	53.12 |
> | Hint | 67.95|	60.84|	60.34|	52.76 |
> | CRD | 61.66|	63.57|	55.31|	53.12 |
>
> | *PACS* | Photo (unseen) |	Sketch (seen)|	Cartoon (seen)|	Art (seen) |
> |-----------|:-----------:|:-----------:|:-----------:|:-----------:|
> | Independent | 40.91|	70.17|	66.47|	45.56 |
> | KL |  49.93|	72.82|	69.69|	46.64|
> | Hint | 48.34|	71.13|	68.08|	45.89 |
> | CRD | 44.49|	71.52|	67.14|	47.86 |
>
>
>
>
> **References**
>
> [1] ImageNet-trained CNNs are biased towards texture; increasing shape bias improves accuracy and robustness. Geirhos et al. ICLR 2019.
>
> [2] Explaining and Harnessing Adversarial Examples. Goodfellow et al. arXiv 2015
>
> [3] Learning Deep Features for Discriminative Localization. Zhou et al. CVPR 2016.
>
> [4] Deeper, Broader and Artier Domain Generalization. Li et al. ICCV 2017.

---

> ### Author Response · Authors · 2023-08-15
>
> Dear reviewer, We appreciate your thorough review and constructive feedback. Our additional experiments are designed to strengthen the evidence supporting the generalizability of our findings across different datasets and distillation objectives. Please let us know if there are any further questions that need clarification.

---

> > ### Comment · Reviewer_B1Ad · 2023-08-17
> > **Rebuttal acknowledgement**
> >
> > I thank the authors for the rebuttal. The added results resolve my concerns and I have increased my rating from 5 to 6.

---

### Official Review · Reviewer_QC8K · 2023-07-08

**Soundness:** 2 fair
**Presentation:** 3 good
**Contribution:** 3 good
**Rating:** 5
**Confidence:** 4

**Summary:**

The paper designed a series of experiments to fundamentally dissect and understand knowledge distillation. The key questions include, Does localization knowledge get distilled? Does adversarial vulnerability get distilled? Does invariance to data transformations get distilled? Does knowledge about unseen domains get distilled? Can students inherit harmful biases from the teacher? and the main content of this paper is to design experiments to answer these questions empirically. At last, the paper provides some intuition about why knowledge distillation works in this way, and concludes with open questions on how to leverage such findings.


**Strengths:**

The paper proposed to comprehensively study the critical, interesting and fundamental problem about deep understanding of knowledge distillation, from an angle of what properties of the teacher are distilled into students, which is a novel angle.

The paper extracted the key questions in order to understand knowledge distillation, where those questions are interesting and essential to help the problem better. Also, for each question, the paper designed corresponding experiments on the representative models with standard datasets.

The paper is easy to follow.


**Weaknesses:**

Even though the paper proposed to study a critical fundamental problem, the paper focused more on designing the experiments to answer the potential questions that could help understand the questions. However, many answers towards the questions are not that surprising, which falls in the expectation. Purely relying on some empirical results to facilitate the understanding of the questions limits the contribution of this paper. If the paper could have some theoretical results to demonstrate why it works and when it works, it could solidify the contribution of this paper.

For each experiment, it’s unclear whether the conclusion could still hold for various architectures or datasets. So it would be great that the paper could include more representative architectures and broader datasets to make results more convincing.

The contributions of this paper would be more solid if there are some attempts to answer some of the open questions in the conclusion part, e.g. how to design algorithms to control which properties to distill or not distill.

**Questions:**

The intuition presented in section 6 could be formalized lowering variance as in the following paper?
Menon et al, “A Statistical Perspective on Distillation”, ICML 2021.


**Limitations:**

yes

---

> ### Author Rebuttal · Authors · 2023-08-09
>
> **1. However, many answers towards the questions are not that surprising.**
>
> While we agree that an observation pertaining to knowledge distillation being surprising could be rather subjective, we want to reiterate the context in which one should look at our results. As explained in lines 30-36, there have been popular works which have casted doubt if knowledge distillation is really working at all, if it cannot improve the accuracy of the student or if it fails to improve the student and teacher's agreement on class predictions. And because of those suspicions, we believe that it *is* rather surprising that so much can happen when a student is being trained to mimic a teacher, even when the student's accuracy doesn't seem to improve (please see Sec. 3.1 in the appendix).
>
>
> **2. For each experiment, it’s unclear whether the conclusion could still hold for various architectures or datasets. So it would be great that the paper could include more representative architectures and broader datasets to make results more convincing.**
>
> In the main paper, we have presented results using various combinations of neural networks for teacher and student models. A list of architectures explored:
> 1. Convolutional Neural Networks
> - ResNet50
> - ResNet18
> - ResNet20
> - VGG19
> - VGG11
> - We investigate simplistic CNN architectures as well (MNIST setting; Sec. 5.1)
> 2. Vision Transformers
> - ViT (ViT-b-32)
> - Swin-Base and Swin-Tiny
>
> Additionally, we have included experiments involving different types of CNNs in our rebuttal.
> - Wide-ResNets (WRN-40-2)
> - ShuffleNets (ShuffleNetV1)
> - MobileNets (MobileNetV2)
>
> While it is true that due to computational and page limit constraints, we could not study all the properties for every combination of architectures (please see our response to reviewer **r3A2**’s point about adding a weakness section), our goal in this work has simply been to show the possibility of the transferability of many properties from a teacher into a student network.
>
> Furthermore, we report results on some additional datasets as well (CIFAR-100, VLCS and PACS); please see our response to reviewer **B1Ad**.
>
> **3. The intuition presented in section 6 could be formalized lowering variance as in the following paper? Menon et al, “A Statistical Perspective on Distillation”, ICML 2021**
>
> We thank the reviewer for bringing our attention to the referenced work. The paper effectively argues that using one-hot labels in the training objective significantly reduces granularity. Moreover, the paper provides a mathematical proof establishing that matching the true probabilities derived from the original data distribution results in a noteworthy reduction in the variance of empirical risk, thereby resulting in improved generalization. Additionally the paper shows that a good teacher must be capable of offering an accurate approximation of these true probabilities.
>
> Nonetheless, we wish to underscore the primary objective behind the intuition presented in Section 6 of our manuscript. Rather than aiming to showcase the utility of knowledge distillation in bolstering generalization, this section primarily aims to elucidate a distinct phenomenon: the potential to reconstruct the precise decision boundary of a teacher even when operating with a limited subset of data points. We furthermore extend this to empirically show that the decision boundary is well approximated even in the cases of deep neural networks. This, we believe, will aid the reader in developing an intuitive understanding of why certain implicit properties are transferred.
>
> That being said, we will include a discussion of this work in the camera ready into a section which deals with different ways in which people have tried to explain why knowledge distillation helps improve the student's performance.

---

> > ### Comment · Reviewer_QC8K · 2023-08-17
> > **thanks**
> >
> > thanks for your detailed explanation. I have increased my rating.

---

> ### Author Response · Authors · 2023-08-15
>
> Dear reviewer, we would like to thank you for the insightful feedback. We hope that our experiments demonstrate the generalizability of our findings to various datasets and other types of architectures as well. Please let us know if there are any other doubts that we could help clarify.

---

### Author Rebuttal · Authors · 2023-08-09

We thank the reviewers for their time in giving us constructive feedback. It is encouraging to learn that the reviewers found our work to be an interesting investigation into a fundamental problem concerning knowledge distillation (QC8K, 8Y8m) using a novel angle (QC8K, r3A2, 8Y8m). The reviewers found the experiments to be well organized (B1Ad, r3A2, 8Y8m) studying the topic comprehensively (QC8K) by covering many kinds of distillation objectives and architectures (8Y8m, r3A2) and investigative different types of properties (8Y8m). Overall, the reviewers found the presentation easy to follow with insightful discussions (QC8K, B1Ad), having interesting findings about a consistent behavior of knowledge transfer from teacher to the student (r3A2, B1Ad, 8Y8m).

A common point raised by many of the reviewers is that the results will be more convincing if this topic is studied on datasets/settings other than the classification task on ImageNet. We have conducted some experiments accordingly, to show that the implicit transfer of properties from the teacher into a student is a rather universal phenomena spanning across a range of tasks/datasets. We address the specific points raised by the reviewers below.

---

### Decision · Program_Chairs · 2023-09-21

**Decision:**

Accept (poster)

**Comment:**

The paper conducts an empirical study of behaviors that are being implicitly transferred through knowledge distillation (KD) from a teacher model to the student, even though these are not explicitly matched by the distillation objective. The paper includes a series of experiments for analyzing the nature of the knowledge "distilled", showing that various KD methods transfer implicit properties, such as localization information, adversarial vulnerabilities, invariance to data transformations and domain adaptation.

The general concern of the reviewers was regarding whether the findings in this work would generalize to other datasets and models. The authors have addressed this in the rebuttal to an extent that the reviewers found reasonable. In addition, multiple reviewers raised concerns about the results not being entirely surprising or exciting, and the empirical nature of this study that does not provide any theoretical support for the findings.

All these points were thoroughly discussed with the reviewers and, overall, there was an agreement that the experiments in the paper are solid and sound, and the findings are novel. Moreover, the above concern seems to be not so critical given the empirical nature of this field, and is overshadowed by the potential of this work to spark interest in the community to better understand the inner workings of KD.